# Kv1.1 contributes to a rapid homeostatic plasticity of intrinsic excitability in CA1 pyramidal neurons in vivo

Peter James Morgan[1]*, Romain Bourboulou[1], Caroline Filippi[1],
Julie Koenig-Gambini[1,2], Jérôme Epsztein[1]*

[1]Institute of Neurobiology of the Mediterranean Sea (INMED), Turing Center for Living Systems (CENTURI), Aix-Marseille University, INSERM, Marseille, France; [2]Institut Universitaire de France, Paris, France

**Abstract** In area CA1 of the hippocampus, the selection of place cells to represent a new environment is biased towards neurons with higher excitability. However, different environments are represented by orthogonal cell ensembles, suggesting that regulatory mechanisms exist. Activity-dependent plasticity of intrinsic excitability, as observed in vitro, is an attractive candidate. Here, using whole-cell patch-clamp recordings of CA1 pyramidal neurons in anesthetized rats, we have examined how inducing theta-bursts of action potentials affects their intrinsic excitability over time. We observed a long-lasting, homeostatic depression of intrinsic excitability which commenced within minutes, and, in contrast to in vitro observations, was not mediated by dendritic $I_h$. Instead, it was attenuated by the Kv1.1 channel blocker dendrotoxin K, suggesting an axonal origin. Analysis of place cells' out-of-field firing in mice navigating in virtual reality further revealed an experience-dependent reduction consistent with decreased excitability. We propose that this mechanism could reduce memory interference.

**\*For correspondence:**
peter.morgan@inserm.fr (PJM);
jerome.epsztein@inserm.fr (JE)

**Competing interests:** The authors declare that no competing interests exist.

## Introduction

The contribution of intrinsic excitability to learning has often been overlooked in favor of synaptic mechanisms (*Daoudal and Debanne, 2003*; *Zhang and Linden, 2003*), but recent evidence indicates that it plays an important role in memory allocation processes and the creation of engrams (*Josselyn and Frankland, 2018*; *Titley et al., 2017*; *Lisman et al., 2018*). In the amygdala and hippocampus it has been shown that the recruitment of cells to represent any new association or environment is biased towards those cells with high levels of intrinsic excitability (*Yiu et al., 2014*; *Pignatelli et al., 2019*; *Grosmark and Buzsáki, 2016*; *Alme et al., 2014*; *Rich et al., 2014*; *Buzsáki and Mizuseki, 2014*; *Han et al., 2007*; *Zhou et al., 2009*; *Epsztein et al., 2011*). In particular, intracellular recordings from individual CA1 pyramidal neurons have shown that higher excitability correlates with neurons becoming place cells in a novel environment (*Epsztein et al., 2011*), and that enhancing excitability with depolarizations of just a few millivolts is sufficient to reveal place specific activity in previously silent cells (*Lee et al., 2012*).

Intrinsic excitability can therefore influence memory allocation processes. However, given that the populations representing different environments are largely orthogonal (*Leutgeb et al., 2004*; *Leutgeb et al., 2005*), a regulatory mechanism is needed to diversify recruitment and prevent the same cells from always being activated.

One attractive mechanism is homeostatic regulation of intrinsic excitability based on firing rate history, as it is known that neurons can adapt their excitability to maintain their activity within set bounds (*Turrigiano et al., 1994*; *Desai et al., 1999*). However, these changes occur slowly in response to sustained changes in activity levels rather than activity patterns associated with coding.

Faster changes in intrinsic excitability that accompany synaptic plasticity have also been described, though these are typically synergistic (going in the same direction than the observed synaptic changes) and restricted to the dendritic branches containing the potentiated synapses (*Daoudal and Debanne, 2003*; *Zhang and Linden, 2003*). Some studies have shown, however, that high frequency bursts of action potentials alone can induce a homeostatic decrease in the global excitability of CA1 pyramidal neurons (*Fan et al., 2005*; *Narayanan and Johnston, 2007*). Such global change in excitability could be particularly relevant to regulate the allocation process as it allows cells to tune their excitability based on their involvement in coding, without disrupting information contained within the synapses. However, this plasticity was mediated by an upregulation of the hyperpolarization-activated cation current ($I_h$), and it is thus questionable whether it would serve this function under natural conditions, as $I_h$ is primarily expressed in the dendrites where it strongly influences synaptic integration (*Magee, 1998*; *Tsay et al., 2007*) and plasticity (*Maroso et al., 2016*; *Nolan et al., 2004*). Stimulations inducing weaker, more physiological, levels of synaptic potentiation can also have the opposite effect, locally reducing $I_h$ and enhancing dendritic excitability around potentiated synapses (*Campanac et al., 2008*). A central question, therefore, is whether an activity-dependent plasticity of global excitability is present in CA1 pyramidal cells in vivo.

To date, only a few studies have described intrinsic plasticity in vivo (*Mahon and Charpier, 2012*; *Paz et al., 2009*; *Aou et al., 1992*; *Belmeguenai et al., 2010*), and the underlying mechanisms were not identified. Here we have examined the effect of a theta-burst firing pattern induced directly through the patch pipette on the intrinsic excitability of CA1 pyramidal neurons in vivo. We found a homeostatic decrease in excitability that was superficially similar to an $I_h$-mediated plasticity that has been described in vitro (*Fan et al., 2005*; *Narayanan and Johnston, 2007*). However, mechanistically our effect was independent of $I_h$, and instead our findings implicate an increase in $I_D$ carried by Kv1.1 channels in the axon initial segment (AIS). This indicates that in vivo, action potential firing can regulate global intrinsic excitability independently of dendritic excitability, which may support the co-existence of synaptic learning and memory allocation processes based on plasticity of intrinsic excitability.

## Results

### Theta burst firing induces a homeostatic decrease in intrinsic excitability of CA1 pyramidal neurons in vivo

We obtained whole-cell patch-clamp recordings from CA1 pyramidal neurons (n = 56) in anesthetized rats using the blind technique (*Lee et al., 2009*; *Margrie et al., 2002*). To determine if the burst (100 Hz) firing of CA1 pyramidal cells at theta (5 Hz) frequency, as observed during place field traversals (*Epsztein et al., 2011*; *Harvey et al., 2009*), could induce long-term changes in intrinsic excitability we applied a theta-burst stimulation (TBS) protocol directly through the patch pipette (*Figure 1A*; see methods). We used an EPSP-like waveform for the current pulses as this reliably entrained the cells to fire at 100 Hz, and where failures occurred they were at the beginnings and ends of the bursts. The identity of the cell was verified in 45 out of 56 cases by biocytin staining (*Figure 1B*). Neuronal excitability was assessed by measuring the firing rate of CA1 pyramidal cells in response to a series of depolarizing current steps from a holding potential of −60 mV (with liquid junction potential correction this corresponds to a membrane potential of around −75 mV). The TBS was applied immediately after the first testing protocol, and the testing protocol was then repeated every 10 min.

Following TBS we saw a long-lasting reduction in intrinsic excitability that began within minutes of the stimulation (*Figure 1C,D*). Average firing rates were significantly reduced 20 min post-TBS (Change in firing rate: TBS −29.2 ± 7.1%, p=7.3×10$^{-4}$, n = 16, *Figure 1E*), which was not seen in control cells (−5.3 ± 4.8%, p=0.20, n = 11, *Figure 1F*). The magnitude of the reduction correlated with the number of spikes evoked during the TBS ($R^2$ = 0.40, p=0.0083; *Figure 1G*) in line with a homeostatic mechanism. To see if the gain of the input-output relationship was altered by the change in excitability we first corrected for variations between cells by normalizing the current-firing responses of each cell and then fitted the population with linear regression. We found that the reduction in firing rate after TBS was uniform across all current step amplitudes, resulting in a rightward shift in the input/output curve with no change in the slope of the fits (Slope: Pre vs 10 min

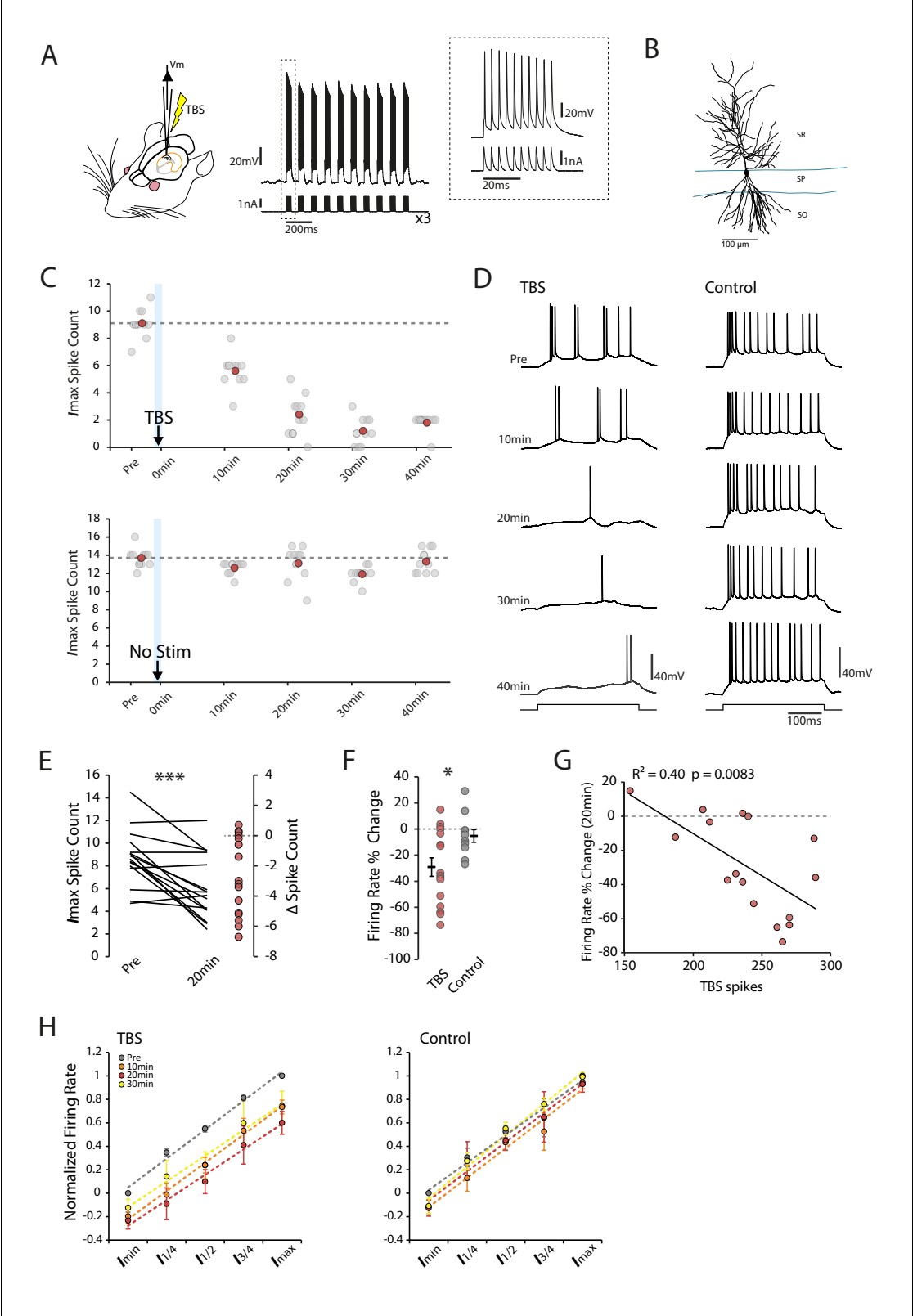

**Figure 1.** Theta-burst stimulation reduces intrinsic excitability. (**A**) Left In vivo whole-cell recordings were made from CA1 pyramidal neurons. The TBS was applied through the patch pipette. Right An example of TBS. EPSP-like waveforms were applied in bursts of 10 (at 100Hz, inset), these bursts were repeated 10 times at theta frequency (5Hz) and the process was repeated three times with an interval of 15 seconds. (**B**) Morphology of a recorded cell. SR stratum radiatum, SP stratum pyramidale, SO stratum oriens. (**C-D**) Examples of a TBS and a control cell. The plots show the response to each of the

*Figure 1 continued on next page*

*Figure 1 continued*

maximum current injections at different times during the testing protocol (grey circles). The red circles mark the mean response at each timepoint. The traces shown in (D) are marked by dark grey borders at their respective timepoints. (E) Left Mean number of spikes evoked in response to the maximal current injection prior to and 20 minutes post-TBS for each cell. Right Distribution of changes in spike count (F) Comparison of change in firing rate at 20 minutes between TBS and control groups. (G) Correlation between the number of spikes evoked during TBS and the change in firing rate. (H) Firing rate vs injected current for the populations of TBS and control cells. The current range is normalized to correct for differences in the range of current amplitudes used. In the TBS group there was a significant reduction (p < 0.05) in the intercept at 10 min (green) and 20 min (red) post-TBS but no change in the gain of the fits. Error bars show mean ± SEM. TBS n = 16 (30 min n = 14), Control n = 11 (30 min n = 6).

The online version of this article includes the following source data and figure supplement(s) for figure 1:

**Source data 1.** Spike count in TBS and control conditions.

**Figure supplement 1.** Change in excitability is not related to inital state of the cells.

p=0.78, 20 min p=0.34 and 30 min p=0.43, *Figure 1H*). As a result the current threshold was significantly increased by the TBS protocol (Intercept: Pre vs 10 min p=0.044, n = 16, 20 min p=0.024, n = 16, 30 min p=0.28 n=14). No effect on slope or intercept was seen in control cells (Pre vs 10 min p=0.26, n = 11, 20 min p=0.49, n = 11, 30 min p=0.27, n = 6, *Figure 1H*). The effect of TBS did not appear to relate to the initial excitability state of the cell, as the change in firing rate did not show any relationship with the initial input resistance ($R^2$ = 0.11, p=0.21, F *Figure 1—figure supplement 1A*), membrane potential ($R^2$ = 0.015, p=0.66, *Figure 1—figure supplement 1B*), action potential threshold ($R^2$ = 0.0045, p=0.80, *Figure 1—figure supplement 1C*) or burstiness ($R^2$ = 0.02, p=0.60, *Figure 1—figure supplement 1D*). Our results therefore show that a short episode of theta-burst firing is sufficient to induce a fast and long-lasting decrease in intrinsic excitability of CA1 pyramidal cells in vivo.

## Plasticity of intrinsic excitability is associated with changes in Rin and first spike latency

We next looked for changes in cellular properties that could indicate what mechanisms might explain the reduction in excitability. We found a reduction in the input resistance following TBS that correlated well with the reduction in excitability (pre vs 20 min p=0.0087, n = 16; $R^2$ = 0.38, p=0.012, *Figure 2A–C*), and that was significant compared to control cells (Change in Rin: TBS −9.7 ± 2.6%, n = 16, Control −1.8 ± 2.5%, n = 11, p=0.043, *Figure 2D*). We similarly found an increase in the latency to first spike that correlated with the change in excitability (pre vs 20 min p=0.005, n = 16; $R^2$ = 0.69, p=1.6×$10^{-4}$, *Figure 2E–G*), and that was significant compared to control (Change in latency: TBS 103.4 ± 29.5%, n = 16, Control 19.6 ± 12.8%, n = 11, p=0.034, *Figure 2H*). These observations suggest that sub-threshold conductances might be modified by the TBS protocol.

To verify that the synaptic drive on to the recorded neurons remained constant for the duration of the recording we compared Vm variance and amplitude before and after application of the TBS protocol. There was no difference in variance before versus 20 min post-TBS (pre vs 20 min p=0.84, n = 16, *Figure 2—figure supplement 1A–B*), no correlation with the change in excitability ($R^2$ = 0.03 p=0.53, *Figure 2—figure supplement 1C*) or difference compared to control cells (ΔVariance: TBS −0.03 ± 0.16 mV$^2$ n=16, Control 0.39 ± 0.22 mV$^2$, n = 11, p=0.12, *Figure 2—figure supplement 1D*).

We also looked for changes in the action potential threshold and spike afterhyperpolarization (AHP), properties which have been associated with changes in excitability in vivo and ex vivo, respectively (*Mahon and Charpier, 2012*; *Oh et al., 2003*). Although after TBS there was an increase in the action potential threshold (pre vs 20 min p<0.019, n = 16, *Figure 2—figure supplement 2A*) the difference between control and TBS cells was not significant (ΔThreshold: TBS 1.16 ± 0.44 mV, n = 16, Control 0.44 ± 0.54 mV, n = 10, p=0.32, *Figure 2—figure supplement 2B*). We did not find any effect on the AHP (pre vs 20 min p=0.92, ΔAHP: TBS 0.4 ± 0.48 mV, n = 16, Control −0.48 ± 0.37 mV, p=0.19, n = 11, *Figure 2—figure supplement 2C,D*). These observations indicate that changes in supra-threshold conductances are unlikely to explain the decrease in excitability we observed.

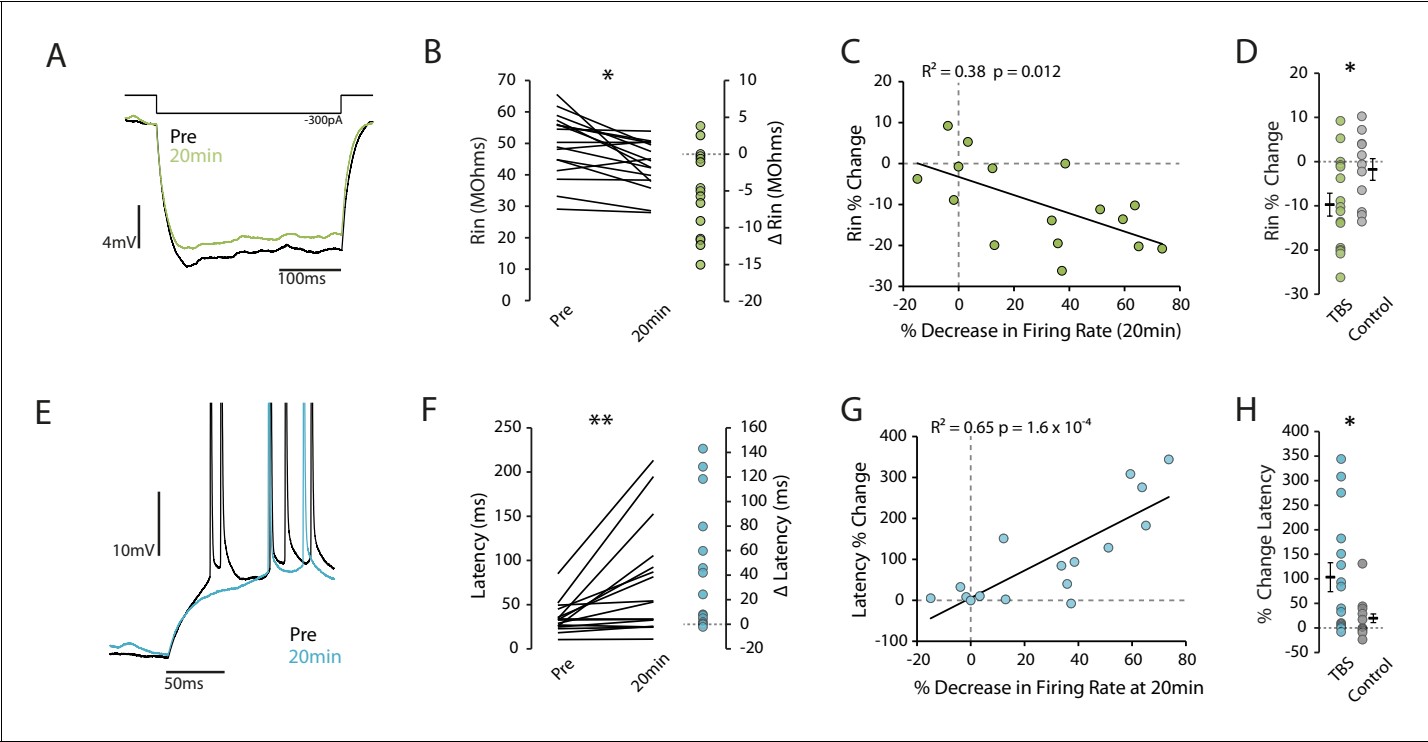

**Figure 2.** Reduction in excitability is associated with decreased Rin and increased latency to first spike. (**A**) Example traces showing a reduction in input resistance. Each trace is the mean of 10 traces. Rin was calculated on hyperpolarizing steps only and the voltage was taken as the minimum within the first 150ms. (**B**) Input resistance prior to and 20 minutes post-TBS. (**C**) Correlation between the change in input resistance and the change in firing rate. (**D**) Comparison of change in input resistance at 20 minutes between TBS and control groups. (**E**) Example trace showing increase in latency after TBS. (**F**) Latency to first spike prior to and 20 minutes post-TBS. (**G**) Correlation between the change in latency and the change in firing rate. (**H**) Comparison of change in latency at 20 minutes between TBS and control groups. Error bars show mean ± SEM. TBS n = 16, Control n = 11.

The online version of this article includes the following source data and figure supplement(s) for figure 2:

**Source data 1.** Input resistance and first spike latency in TBS and control conditions.
**Figure supplement 1.** TBS did not change background synaptic activity.
**Figure supplement 2.** Action potential threshold and medium AHP.

## Plasticity induction depends on rise in intracellular calcium but is independent of NMDARs

If the plasticity we observed indeed depends on action potential firing, we would expect it to be dependent on increases in intracellular calcium. To verify that this was the case we repeated the experiments with the addition of the calcium chelator BAPTA (0.5 mM) to the intracellular solution. As expected, in the presence of BAPTA the depression of intrinsic excitability was prevented (BAPTA n = 7, change in firing rate: 23.0 ± 13.7%, p=0.0012 vs TBS; change in Rin: 2.4% ± 5.16, p=0.029 vs TBS; change in latency: 8.7 ± 7.4%, p=0.05 vs TBS, *Figure 3*).

We next blocked calcium influx through NMDARs, as this has been described to block various mechanisms for changes in intrinsic excitability in vitro (*Fan et al., 2005*; *Wang et al., 2003*; *Frick et al., 2004*; *Losonczy et al., 2008*), even in the absence of synaptic stimulation (*Fan et al., 2005*). In the latter case it is assumed that there is either background NMDAR activity or that the backpropagating action potentials depolarize the dendrite sufficiently to remove the $Mg^{2+}$ block for ongoing activity to activate NMDARs. We found, however, that blocking NMDARs by adding 1 mM MK-801 to intracellular solution did not affect the TBS-induced reduction in excitability (MK-801 n = 7, change in firing rate: −42.1 ± 12.2%, p=0.35 vs TBS; change in Rin: −13.2 ± 2.5%, p=0.43 vs TBS; change in latency: 156.4 ± 70.3%, p=0.42 vs TBS, *Figure 4*). This is perhaps unsurprising given the lack of potentiating synaptic stimulation, and we can infer that influx through voltage-gated

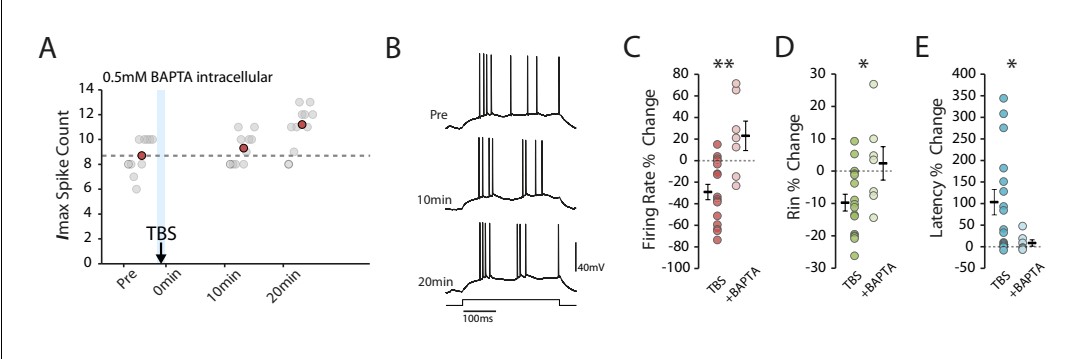

**Figure 3.** Calcium chelation prevents the TBS-induced decrease in excitability. (**A-B**) Example cell showing that the decrease in excitability is blocked with 0.5mM BAPTA inside the patch pipette. Scatter plots show the response to each of the maximum current injections during the testing protocol (grey circles). The red circles mark the mean response at each timepoint. Dark grey borders mark the current steps depicted in (**B**). (**C-E**) Comparisons of changes in firing rate (**C**), input resistance (**D**) and latency (**E**) between TBS alone (n = 16) and TBS with BAPTA (n = 7). Error bars show mean ± SEM. The online version of this article includes the following source data for figure 3:

**Source data 1.** Spike count, input resistance and first spike latency in TBS condition with BAPTA.

calcium channels is the major, or at least initial, source of calcium for the induction of the plasticity we observed.

## Decreased excitability cannot be explained by an increase in $I_h$

Our observations to this point are consistent with the $I_h$-mediated reduction in intrinsic excitability seen in vitro (*Fan et al., 2005*; *Campanac et al., 2008*). To see if $I_h$ could similarly explain the reduction in excitability, we looked to see if TBS affected the sag ratio (a measure of the voltage change resulting from $I_h$ activation, *Figure 5A*). However, we found that the sag ratio actually increased following TBS (pre vs 20 min p=0.0088, n = 16, *Figure 5B*), which indicates a decrease in $I_h$ that would be expected to make the cells more excitable. This decrease was not detected in the presence of BAPTA and MK-801 (BAPTA p=0.39, n = 7, MK-801 p=0.97, n = 7, *Figure 5C*), conditions which blocked the in vitro effects of TBS on $I_h$ (*Fan et al., 2005*). The change in sag ratio was inversely correlated with the strength of stimulation ($R^2$ = 0.42, p=0.0065, *Figure 5D*), which is consistent with

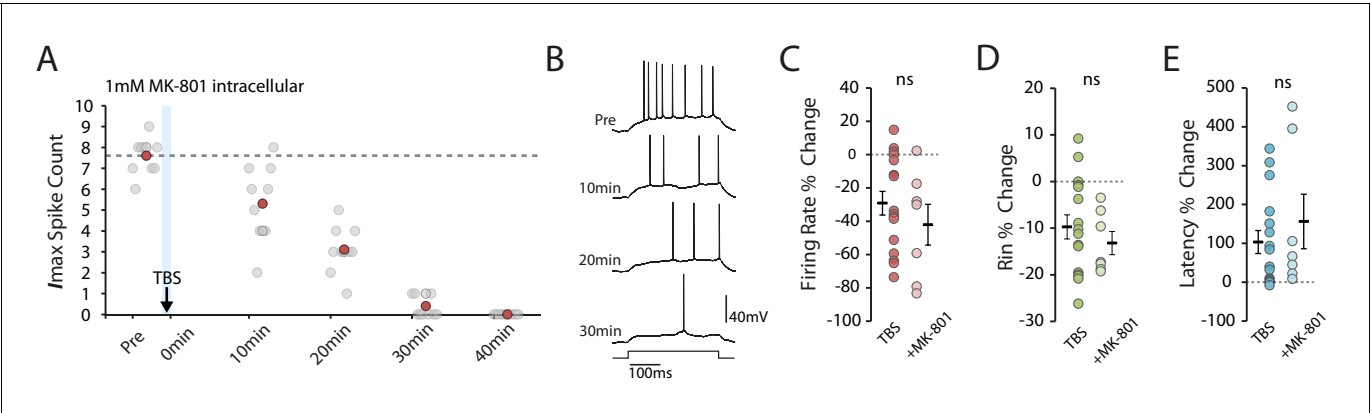

**Figure 4.** NMDAR block does not prevent the TBS-induced decrease in excitability. (**A-B**) Example cell showing that 1mM MK-801 inside the pipette does not prevent the decrease in excitability. Scatter plot shows the response to each of the maximum current injections during the testing protocol (grey circles). The red circles mark the mean response at each timepoint. Dark grey borders mark the current steps depicted in (**B**). (**C-E**) Comparisons of changes in firing rate (**C**), input resistance (**D**) and latency (**E**) between TBS alone (n = 16) and TBS with MK-801 (n = 7). Error bars show mean ± SEM. The online version of this article includes the following source data for figure 4:

**Source data 1.** Spike count, input resistance and first spike latency in TBS condition with MK801.

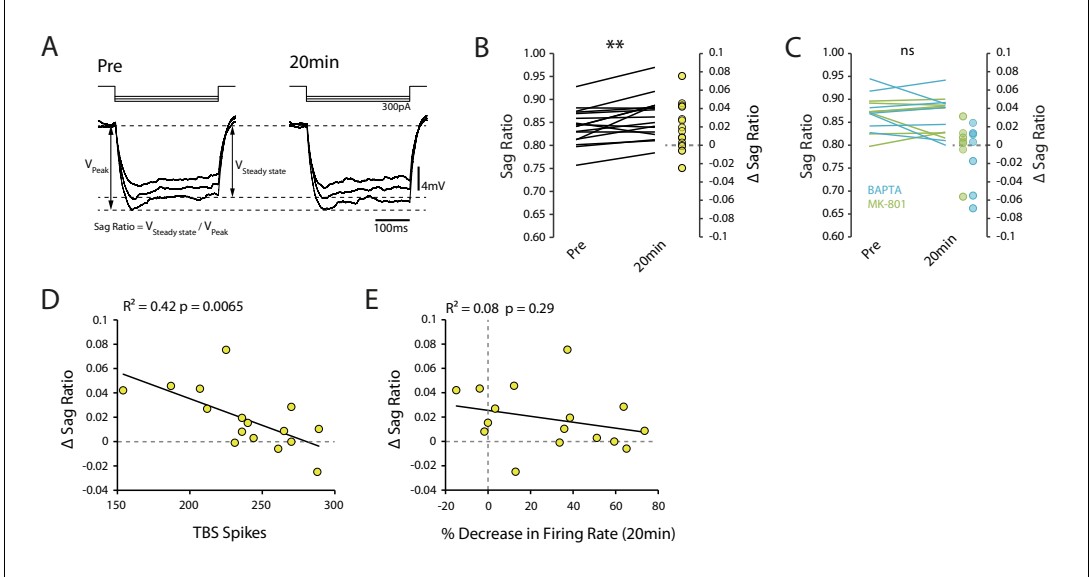

**Figure 5.** Changes in $I_h$ do not account for decrease in excitability. (**A**) Example traces showing the voltage sag caused by $I_h$ pre and 20 min post-TBS. Each trace is the mean of 10 trials. (**B**) Sag ratio prior to and 20 minutes post-TBS (n = 16). (**C**) Sag ratio prior to and 20 minutes post-TBS with 0.5mM BAPTA (n = 7) or 1mM MK-801 (n = 7) in the patch pipette. (**D**) Correlation between the change in sag ratio and the number of spikes evoked during TBS. (**E**) Scatter plot showing that there was no strong relationship between the change in sag ratio and the decrease in firing rate.

The online version of this article includes the following source data for figure 5:

**Source data 1.** Sag ratio in TBS condition without drugs and with BAPTA and MK801.

previous observations following weak LTP induction (*Campanac et al., 2008*), but did not significantly correlate with the overall change in excitability ($R^2$ = 0.08, p=0.29, *Figure 5E*). To more directly assess the role of $I_h$ in the plasticity we observed, we repeated the experiments with the addition of 100 µM ZD7288 (a potent and specific blocker of $I_h$ channel with an intracellular site of action) to the intracellular solution. Under this condition the membrane sag was abolished and Rin was increased (Pre-TBS Rin: TBS 48.7 ± 2.5 MΩ, ZD 68.6 ± 5.9 MΩ), indicating $I_h$ was blocked (*Figure 6A–B*). However, intracellular ZD7288 did not prevent TBS-induced decrease in firing rate (ZD n = 6, change in firing rate: −22.9 ± 8.6%, p=0.63 vs TBS, *Figure 6C–E*), decrease in input resistance (ZD change in Rin: −14.3 ± 3.8%, p=0.36 vs TBS, *Figure 6F*) and increase in first spike latency (ZD change in latency: 78.7 ± 13.1%, p=0.62 vs TBS, *Figure 6G*).

These results indicate that while plasticity of $I_h$ might have been induced by our stimulation protocol, it would tend to increase rather than decrease intrinsic excitability. Therefore plasticity in another, opposing, mechanism determined the overall change in excitability.

## $I_D$ contributes to the decrease in intrinsic excitability

Failing a role for $I_h$, one possible source for the decreased excitability would be potassium conductances. The large increases in first spike latency are suggestive of $I_D$, a slowly inactivating potassium current activated at subthreshold membrane potentials (*Storm, 1988*), which in CA1 pyramidal neurons is carried by Kv1.1 channels expressed at the AIS (*Kirizs et al., 2014*). To determine if $I_D$ contributed to the reduction in excitability we applied dendrotoxin K, a Kv1.1 specific channel blocker (*Robertson et al., 1996*). Prior to patching, we pressure injected 200 nM DTX-K for 5 min approximately 100 µm above the pyramidal cell layer, into the region containing the AIS. We then patched cells with an additional 10 nM DTX-K in the pipette solution. In the presence of DTX-K the effect of TBS on excitability appeared to be reduced (*Figure 7A–B*). The TBS protocol induced more spikes in the presence of DTX-K (>250 spikes in 2/3 of the cells in DTX-K but only 1/3 of the cells in control condition). As the effect on excitability depended on the number of spikes induced during TBS (see *Figure 1G*), we focused on those cells where TBS evoked at least 250 spikes in both control and DTX-K conditions. We found that the reduction in firing rate was significantly smaller with DTX-K (change in firing rate: TBS −51.8 ± 8.7%, n = 6, DTX-K −25.5 ± 5.3%, n = 6, p=0.037, *Figure 7C*),

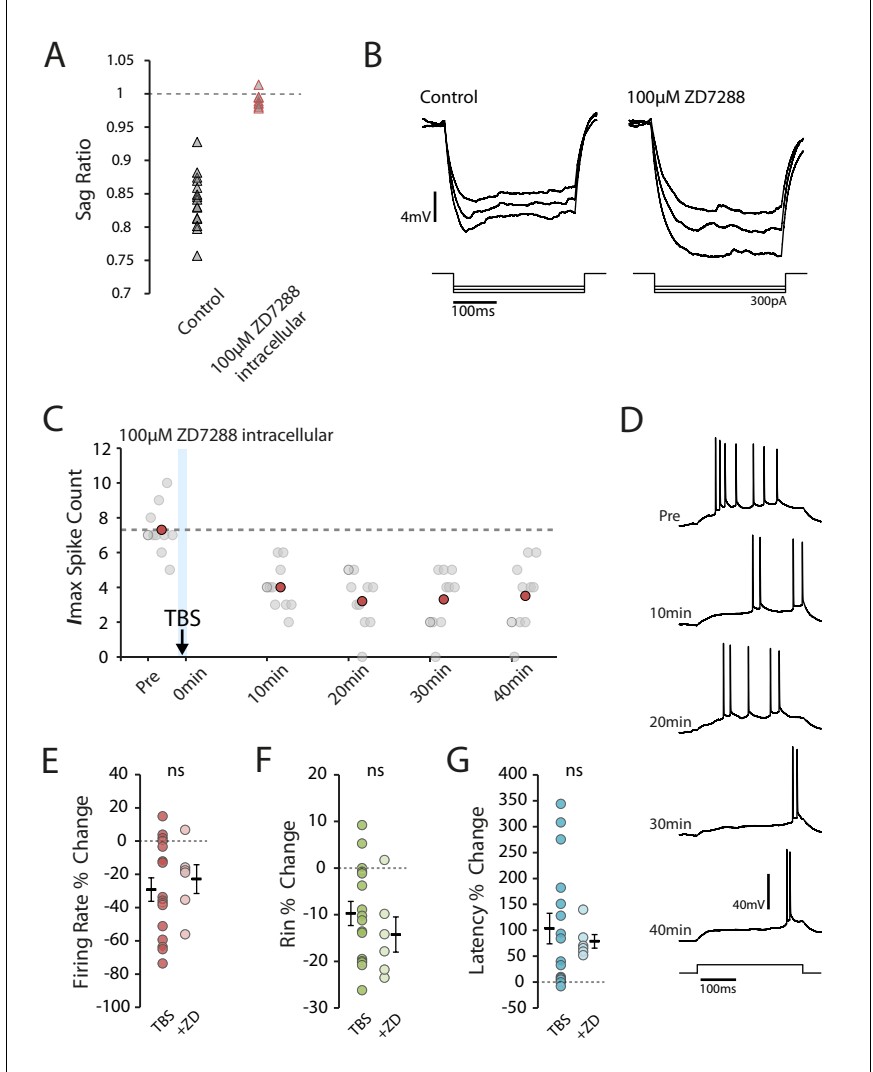

**Figure 6.** Block of $I_h$ does not prevent reduction in excitability. (**A-B**) Inclusion of ZD7288 in the patch pipette blocked $I_h$, abolishing the voltage sag. Each trace shown is the mean of 10 responses. (**C-D**) Example cell showing the effect of TBS in the presence of ZD7288 inside the patch pipette. Scatter plot shows the response to each of the maximum current injections during the testing protocol (grey circles). The red circles mark the mean response at each timepoint. Dark grey borders mark the current steps depicted in (**D**). (**E-F**) Comparisons of changes in firing rate (**E**), input resistance (**F**) and latency (**G**) between TBS alone (n = 16) and TBS with ZD7288 (n = 6). Error bars show mean ± SEM.

The online version of this article includes the following source data for figure 6:

**Source data 1.** Spike count, input resistance and first spike latency in TBS condition with ZD7288.

the reduction in input resistance was mostly abolished (change in Rin: TBS −17.4 ± 1.7%, n = 6, DTX-K −5.5 ± 1.7%, n = 6, p=9.7×10$^{-4}$, **Figure 7D**), as was the increase in latency (change in latency: TBS 192.0 ± 54.2%, n = 6, DTX-K 51.0 ± 10.8%, n = 6, p=0.04, **Figure 7E**). These results suggest that the reduction in excitability following TBS is mediated, at least in part, by an increase in $I_D$.

## Bias towards reduced excitability of place cells during virtual spatial navigation

Our results demonstrate that action potential firing can induce a decrease in the intrinsic excitability of CA1 pyramidal neurons, but the question remains as to whether this plasticity affects coding in

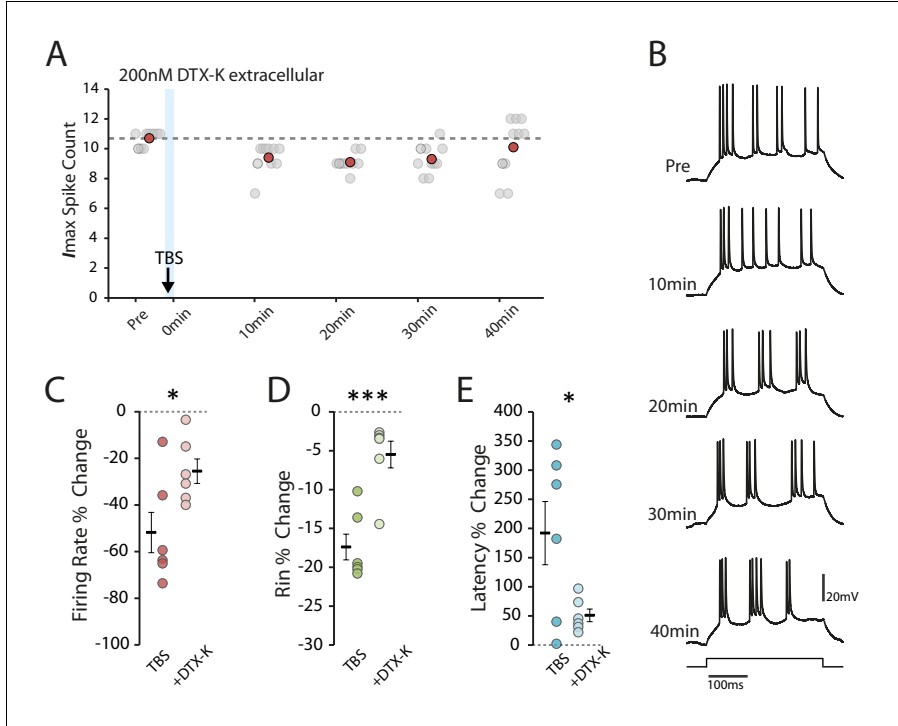

**Figure 7.** Dendrotoxin-K reduces the effect of TBS on excitability. (**A-B**) Example cell recorded after the application of DTX-K. Scatter plot shows the response to each of the maximum current injections during the testing protocol (grey circles). The red circles mark the mean response at each timepoint. Dark grey borders mark the current steps depicted in (**B**). (**C-E**) Comparisons of changes in firing rate (**C**), input resistance (**D**) and latency (**E**) between TBS alone (n = 6) and with DTX-K (n = 6) for cells where TBS evoked at least 250 spikes. Error bars show mean ± SEM.

The online version of this article includes the following source data for figure 7:

**Source data 1.** Spike count, input resistance and first spike latency in TBS condition with DTX-K.

awake animals. To look for evidence of its expression we reanalyzed data from a previous study, where we made extracellular recordings from CA1 in mice while they navigated virtual reality tracks in both novel and familiar conditions (*Bourboulou et al., 2019*). The novel and familiar conditions differed by the availability of local visual cues (virtual 3D objects) and population analysis showed that place cell identity and firing locations between the two were uncorrelated, indicating there was a global remapping. To examine how firing rate changed during the session we used two complementary measures: the firing rate during the start, middle and end time periods of the session (which shows the size of the effect), and the lap-by-lap correlation of firing rate over the entire session (which shows the consistency of the effect). We hypothesized that a change in intrinsic excitability should be visible in the out-of-field firing rate, as unlike in-field firing it should not be affected by synaptic plasticity.

We found that in the novel condition the out-of-field firing rate decreased over time (Start 1.58 ± 0.11 Hz, Middle 1.47 ± 0.1 Hz, End 1.37 ± 0.1 Hz, n = 171, F = 5.17 p=0.0065, *Figure 8A–C*), and that the distribution of the lap-by-lap correlation was shifted towards a decrease in activity (n = 171, p=0.0093, *Figure 8D*). This effect is compatible with decreased intrinsic excitability. The in-field firing rate, in contrast, increased across the session (Start 3.63 ± 0.19 Hz, Middle: 3.92 ± 0.21 Hz, End 4.19 ± 0.22 Hz, n = 171, F = 5.02, p=0.0071, *Figure 8E*) and there was a positive shift in the correlation distribution (p=0.048, n = 171, *Figure 8F*). These observations are consistent with synaptic potentiation within the place field (*Bittner et al., 2015*; *Diamantaki et al., 2018*; *Mehta et al., 1997*; *Cohen et al., 2017*). In the familiar condition we did not detect any change in the out-of-field firing rate (Start 1.71 ± 0.1 Hz, Middle: 1.62 ± 0.1 Hz, End 1.74 ± 0.11 Hz, n = 268, F = 1.53, p=0.22, *Figure 8—figure supplement 1A*), though there was a negative shift in the center of the correlation

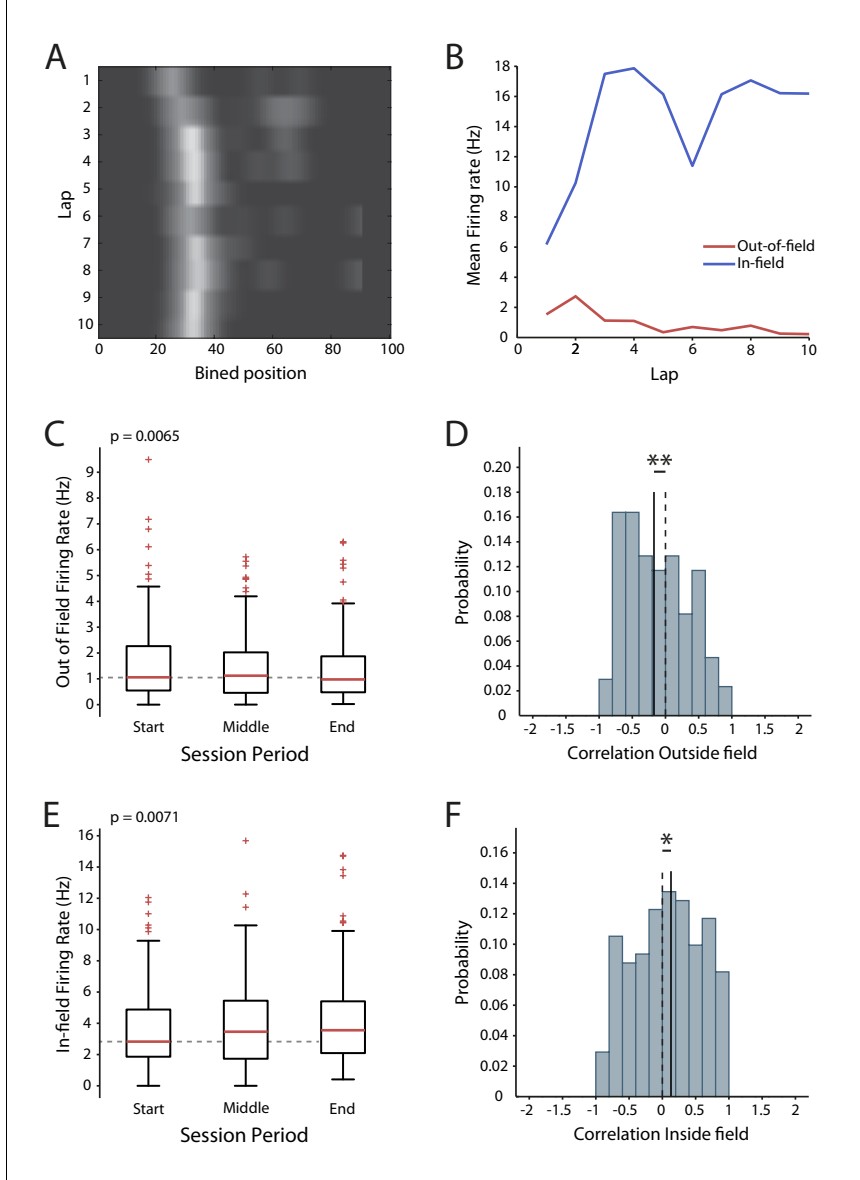

**Figure 8.** Changes in excitability during exploration of a novel virtual reality track. (**A**) Firing rate map of one cell. Lighter colors indicate higher rate. (**B**) In-field and out-of-field firing rate by lap for the example cell. (**C**) Mean firing rates outside the place field across the session. (**D**) Distribution of the out-of-field firing rate correlations. (**E**) Mean in-field firing rates across the session. (**F**) Distribution of the in-field firing rate correlations. Box and whisker plots show median, 25-75th and 5-95th percentiles, n = 171

The online version of this article includes the following source data and figure supplement(s) for figure 8:

**Source data 1.** In-field, out-of-field firing rates and correlations across laps for place cells in new condition.
**Figure supplement 1.** Changes in excitability during exploration of a familiar environment.
**Figure supplement 1—source data 1.** In-field, out-of-field firing rates and correlations across laps for place cells in familiar condition.

(p=0.012, n = 268, *Figure 8—figure supplement 1B*). The in-field firing rate also did not change over time (Start 5.3 ± 0.23 Hz, Middle: 5.34 ± 0.22 Hz, End 5.15 ± 0.22 Hz, n = 268, F = 0.87, p=0.42, *Figure 8—figure supplement 1C*) and there was no change in the firing rate correlation distribution (p=0.25, n = 268, *Figure 8—figure supplement 1D*). Overall, these results are consistent with the presence of an activity-dependent reduction in intrinsic excitability during exploration of a novel environment.

# Discussion

Our results show that short periods of theta-burst firing, mimicking the firing of place cells within their place fields (*Epsztein et al., 2011*; *Harvey et al., 2009*), can induce a global and long-lasting depression of intrinsic excitability in CA1 pyramidal neurons in vivo. This plasticity was homeostatic as its magnitude correlated with the number of action potentials used during the induction protocol. Its induction required intracellular calcium and its expression was partially blocked by the Kv1.1 blocker DTX-K, implicating $I_D$ in the AIS. These observations demonstrate an additional form of plasticity of intrinsic excitability not previously described in CA1 pyramidal neurons in vitro. Furthermore, analysis of place cells in behaving mice revealed a bias towards reduced excitability that is consistent with the expression of this plasticity during spatial navigation.

Previous report of plasticity of global intrinsic excitability of CA1 pyramidal cells in vitro have implicated an upregulation of the dendritic conductance $I_h$ (*Fan et al., 2005*; *Campanac et al., 2008*). However, subsequent work revealed that the direction and spatial extend of $I_h$ modulation depends on the strength of the stimulation used to induce the plasticity. Thus $I_h$ could be locally decreased (thus enhancing intrinsic excitability) in response to more moderate stimulations associated with lower levels of synaptic potentiation (*Campanac et al., 2008*). The change in $I_h$ we observed was small and went in the opposite direction to the overall decrease in excitability, consistent with in vitro observations following moderate stimulation (*Campanac et al., 2008*), and indicates action potential firing only weakly engages $I_h$ plasticity in vivo. One reason for this may be the lower input resistance of dendrites in vivo reducing action potential backpropagation (*Kamondi et al., 1998*; but see *Waters and Helmchen, 2004*) and concomitant calcium entry (*Campanac et al., 2008*). This lower dendritic input resistance in vivo (*Destexhe et al., 2003*), however, also limits the ability of dendritic currents to propagate to the soma, and therefore changes may have occurred in the distal dendrites but had little to no impact on somatic excitability. In either case our results suggest that $I_h$ regulates excitability locally rather than globally under our conditions.

Our results instead suggest that an increase in $I_D$ was contributing to the decrease in excitability, through increasing the delay to the onset of firing. Perhaps surprisingly, we also saw a reduction in the change in input resistance by DTX-K, which suggests that some Kv1 channels are active at the holding membrane potential. This could be explained by the overlap of the activation and de-inactivation voltage dependencies of $I_D$ at around $-75$ mV (*Storm, 1988*), approximately the true holding potential in our recordings, which could allow for a small, continuous $I_D$. This observation also favors a shift in the activation properties of the channels over membrane insertion as the mechanism of regulation, as most of those newly inserted channels would remain closed. In CA1 pyramidal neurons $I_D$ is primarily carried by Kv1.1 and its expression is highly localized to the AIS (*Kirizs et al., 2014*). As this is the site of action potential generation, changes in AIS excitability can have a significant influence on the excitability of a neuron without affecting synaptic integration (*Kole and Stuart, 2012*; *Rama et al., 2017*). Functionally, this distinguishes it from plasticity in dendritic conductances such as $I_h$ and Ia, which primarily regulate the input rather than output of the neuron. That DTX-K did not completely prevent the reduction in excitability might be explained by incomplete block of Kv1.1 channels or the involvement of other Kv1 channel subtypes (*Monaghan et al., 2001*), or by plasticity in other conductances such as inwardly rectifying potassium channels (*Kim and Johnston, 2015*) which are active around the resting membrane potential. Another possibility is an increase in the tonic GABA current, for example through a calcium-dependent increase in GABA-A receptors exocytosis such as has been described in cultured hippocampal neurons (*Saliba et al., 2012*). This increase could contribute to the shift in the input-output curve, though it should also speed up the membrane time constant and therefore decrease the latency to firing (*Wlodarczyk et al., 2013*), opposite to what we observed. Homeostatic plasticity of Kv1 has typically been associated with structural changes over hours or days in response to sustained input (*Kuba et al., 2015*; *Cudmore et al., 2010*), and in this sense the plasticity we observed is unusual, as it occurs rapidly and in response to a relatively small stimulus. Such a sensitive mechanism for homeostasis may be necessary in CA1 as pyramidal neurons are characterized by their near silence outside of their active fields, and therefore homeostatic regulation based on sustained changes in firing rates, as seen in the neocortex, is less relevant under normal conditions.

Extrinsic changes might also be a factor. CA1 pyramidal neurons are embedded in a complex local feed-back inhibitory network (*Pouille and Scanziani, 2004*) and it is possible that the TBS-induced burst firing was sufficient to induce plasticity in recurrently connected interneurons. Indeed, theta-burst firing of CA1 pyramidal cell has been shown to induce plasticity at inhibitory synapses in vitro via retrograde endocannabinoid signaling (*Younts et al., 2013*). This plasticity, however, led to a long-term depression at both somatic and dendritic inhibitory synapses and is thus unlikely to account for the decrease in excitability we observed. On the other hand, feed-forward synaptic stimulations (of the Schaffer collaterals) were shown to induce plasticity of intrinsic excitability (in addition to synaptic plasticity) in parvalbumin-expressing basket cells through a mGluR-dependent downregulation of Kv1 channels (*Campanac et al., 2013*). It is unclear, however, if this plasticity could be engaged via feedback activation. Opposite to this prediction, our effect was prevented by intracellular BAPTA and therefore calcium-dependent, unlike the mGluR-mediated plasticity, and the initial input resistance of CA1 pyramidal cells was increased in the presence of DTX-K, opposite of what would be expected from increased synaptic inhibition. Nonetheless, a contribution of inhibitory plasticity via other mechanisms is difficult to fully rule out in vivo given the paucity of efficient intracellular blockers of GABA-A receptors (*Atherton et al., 2016*).

The differences between our findings and those in vitro may relate to the nature of the different preparations. The in vitro slice preparation offers many advantages to study synaptic and intrinsic plasticity such as the ability to perform long-lasting high-quality recordings, good spatial and temporal control of stimulated fibers and the ease to deliver antagonists of ligand- and voltage-gated channels. Studies using this preparation have led to a wealth of data on plasticity of intrinsic excitability that constitute a solid framework to interpret results obtained in vivo. Nevertheless, the in vitro slice preparation remains artificial, the neuronal activity is strongly reduced and the neuromodulatory system is inoperant. The slicing procedure itself is also traumatic for the neurons, invoking stress responses alongside inevitable damage to the axons and dendrites, which results in cellular changes that could affect the expression of plasticity (*Taubenfeld et al., 2002*; *Ho et al., 2004*). Therefore, an important question is to decipher whether and how plasticity of intrinsic neuronal excitability operate in more naturalistic conditions in vivo. Ideally, these recordings should be performed in awake drug free animals. However these are often short lasting (*Epsztein et al., 2011*; *Lee et al., 2009*; *Harvey et al., 2009*; *Epsztein et al., 2010*; *Lee et al., 2006*) and obtaining long-lasting, stable recordings, necessary to measure long-term plasticity, remains difficult. To balance these problems we recorded in anesthetized rats, which affords more stability than awake preparations at the expense of being in an unnatural brain state. Because of this, some considerations should be made in interpreting our results. The anesthetics used will have directly and indirectly affected neuronal excitability, including via an allosteric potentiation of GABA-A receptors and activation of inwardly rectifying potassium channels (*Möhler et al., 2002*; *Nagi and Pineyro, 2014*; *Hein, 2006*), and under anesthesia the levels of neuromodulators that regulate synaptic plasticity (*Palacios-Filardo and Mellor, 2019*; *Dias et al., 2013*) differ compared to awake states. These factors may have influenced the plasticity we observed, and therefore its significance needs to be confirmed in behaving animals.

We did, however, see changes in out-of-field firing during exploration of virtual reality environments that are consistent with an activity-dependent decrease in intrinsic excitability. These effects were present in novel environments, and although we did not detect any changes in the out-of-field firing rate in the familiar environment, the same tendency towards a decrease was still visible in the firing rate correlations. This leaves open the possibility that the same processes underlying the decrease in excitability were present but masked by other changes. One caveat to consider, however, is that it is not known how synaptic strength outside of the place field is affected by exploration and plasticity within the place field itself (*Diamantaki et al., 2018*). And as discussed above, plasticity of interneuronal activity may have also been involved. Nonetheless, despite these open questions, our results show that in a novel environment the excitability of place cells tends to decrease on a timescale that is consistent with the plasticity we have described.

Fast, sensitive homeostatic plasticity of intrinsic excitability could potentially be involved in the regulation of coding in CA1. Indeed, the population of neurons that encode an environment or experience is only a subset of the population that could potentially do so (*Thompson and Best, 1989*), and those cells that are active tend to have higher levels of intrinsic excitability (*Pignatelli et al., 2019*; *Grosmark and Buzsáki, 2016*; *Alme et al., 2014*; *Rich et al., 2014*;

*Buzsáki and Mizuseki, 2014*; *Epsztein et al., 2011*). Furthermore, studies that have enhanced excitability in a subset of cells by overexpressing CREB have shown that this biases cells to participate in the engram (*Kim et al., 2014*; *Sekeres et al., 2010*). As the population of active cells is roughly constant in both CA1 and the amygdala it appears that selection is determined, in part, competitively through inhibition, a process which the more excitable cells are most likely to win (*Trouche et al., 2016*). To prevent the network from becoming dominated by the same sets of neurons this requires mechanisms to modulate excitability, without interfering with learning already contained within the synapses, for which global forms of plasticity of intrinsic excitability appear ideally suited. The activity-driven decrease in excitability we have described might therefore contribute to such a process. In line with this interpretation, it was recently shown that cFos-labelled engram cells show reduced likelihood to encode a novel environment (*Tanaka et al., 2018*). It remains to be determined, though, how these changes relate to the CREB-associated increases in excitability that have been associated with learning (*Lisman et al., 2018*). Enhanced excitability by CREB is associated with a decrease in the AHP (*Lopez de Armentia et al., 2007*; *Yu et al., 2017*), in which we saw no change. This is unsurprising though, as CREB is not activated by action potential firing, but rather requires calcium entry at the synapses (*Deisseroth et al., 1996*). This points towards the wider question of how action potential-driven plasticity interacts with synaptically-driven changes in intrinsic excitability, as the balance between them may determine how the cell contributes to coding future experiences. In particular, it remains to be determined how intrinsic excitability is modulated following place field activity in behaving animals (*Schmidt-Hieber and Nolan, 2017*; *Debanne et al., 2019*), and if it is affected by the plateau potentials that have been shown to induce place field formation (*Bittner et al., 2015*; *Diamantaki et al., 2018*; *Bittner et al., 2017*).

## Materials and methods

### Key resources table

| Reagent type (species) or resource | Designation | Source or reference | Identifiers | Additional information |
|---|---|---|---|---|
| Chemical compound, drug | ZD7288 | Tocris Bioscience | Cat. No. 1000 | |
| Chemical compound, drug | Dendrotoxin K | Sigma Aldrich | D4183 | |
| Chemical compound, drug | MK-801 | Tocris Bioscience | Cat. No. 0924 | |

### Ethics statement

All experiments were approved by the Institut National de la Santé. et de la Recherche Médicale (INSERM) animal care and use committee and authorized by the Ministère de l'Education Nationale de l'Enseignement Superieur et de la Recherche (agreement number 692 02048.02), in accordance with the European community council directives (2010/63/UE).

### Surgery and anesthesia

Male Wistar rats (P25-P35, Charles River) were anesthetized with a combination of fentanyl (7.5 µg/Kg), medetomidine (225 µg/Kg), midazolam (6 mg/Kg). Maintenance doses of one third the initial dose were given every 45 min. The surface of the skull was cleaned and covered with a thin layer of cyanoacrylate glue, and a headplate attached with dental cement. The craniotomies were drilled at −3.5 mm dorsoventral and 2.5 mm lateral from bregma, and were approximately 0.5 mm in diameter. To allow access with the patch pipettes, the dura was dried and a small incision was made using a 26G needle. The ground electrode was placed underneath the skin at the back of the neck and sealed with Kwik-Cast (World Precision Instruments).

### In vivo electrophysiology

Whole-cell patch-clamp recordings were obtained from CA1 pyramidal cells. Before patching we located CA1 using a low-resistance glass electrode filled with ringer solution (consisting of, in mM:

135 NaCl, 5.4 KCl, 1 MgCl2, 1.8 CaCl2 and 5 HEPES, pH 7.3 with NaOH). This was lowered through the cortex until ripples and bursting activity characteristic of CA1 could be detected in the LFP. Typically this was around 2100 µm deep. Patch electrodes (5-10MOhm) were pulled from borosilicate glass (Harvard Apparatus) and backfilled with an intracellular solution consisting of (in mM): 135 K-Gluconate, 4 KCl, 10 HEPES, 2 Mg-ATP, 0.4 GTP, plus 0.2% biocytin. The pH was adjusted to 7.2 with KOH. To prevent the pipette from clogging high pressure (around 800mbar) was applied while it was slowly lowered to the target depth, about 50 µm above the pyramidal cell layer. Once there the pressure was immediately lowered to between 25 and 35mbar. Before searching for cells the craniotomy was filled with 2% agar solution and the pipette capacitance was compensated. After breaking in we allowed 10 min for the cells to stabilize before starting the protocol.

For the pressure injection of DTX-K, a broken patch pipette (tip diameter approximately 10–20 µm) containing 200 nM DTX-K dissolved in ringer solution was lowered to 100 µm above the pyramidal cell layer. Once positioned, DTX-K was injected with gentle pressure (around 15mbar) for 5 min.

Each TBS series consisted of 10 EPSC-like waveforms delivered at 100 Hz, repeated 10x at 5 Hz. This protocol was repeated 3x at 15 s intervals. The testing protocol consisted of 3 hyperpolarizing current injections followed by between 3–6 depolarizing steps, repeated 10 times. Each step was 300 ms long, with a 1.5 s interval. The hyperpolarizing step amplitudes were −300 pA, −250 pA and −200 pA, while the range of depolarizing step amplitudes was adjusted for each cell, depending on their initial excitability. For the TBS and testing protocols the cell was held at −60 mV. The liquid junction potential was calculated at 17 mV between the intracellular and ringer solutions, and was not corrected.

Data were acquired using a NPI ELC-3XS amplifier (NPI Electronics) and digitized with an LIH-1600 (HEKA Electronik). Acquisition protocols were implemented in Patch Master software (HEKA Electronik). Output signals were passed through a digidata 1440A (Molecular Devices) and recorded with axoscope (Molecular Devices). The current signal was filtered with a 50 Hz notch filter (Quest Scientific).

## Histology

After recording the rats were transcardially perfused with 4% PFA. The brains were then left in 4% PFA overnight at 4°C, before washing and storing at 4°C in PBS. To stain for biocytin 50–100 µm thick slices were incubated in PBS containing 1:1000 streptavidin-alexa 594 (Invitrogen) or streptavidin-cyanine 3 (Jackson Immunoresearch), 0.3% triton X-100% and 2% normal goat serum for 48–72 hr at 4°C, and agitated throughout. Slices were then washed in PBS three times for 10 min each at 4°C and under agitation. Images were acquired under a confocal microscope (SP5X, Leica) with a 40x objective and cells were reconstructed in Neurolucida (MBF Bioscience). In total we recovered 45/56 cells (TBS: 12/16, Control: 4/11, ZD: 6/6, MK: 7/7, BAPTA: 7/7, DTX-K: 9/9).

## Reagents and drugs

All reagents and DTX-K were obtained from Sigma-Aldrich. The drugs ZD7288 and MK-801 were obtained from Tocris Biosciences. Fentanyl (Fentadon, Dechra) and medetomidine (Domitor, Orion Pharma) were obtained from Centravet, and midazolam was obtained from Accord Healthcare France.

## Data analysis

Data were analyzed using clampfit (Molecular devices), Matlab (Mathworks), Statistica (Tibco) and Excel (Microsoft). Only cells with an initial $Vm < -55$ mV, Rs <100 MOhms and spike amplitude >40 mV, and which were stable up to completion of the 20 min post-TBS test protocol were included in the analysis.

Current-firing rate curves were normalized using the equation:

$$(X - minPre)/ (maxPre - minPre)$$

where X is the spike count, minPre is the mean spike count in response to the smallest current step prior to TBS and maxPre is the mean spike count for the largest current step prior to TBS.

To calculate the series resistance an exponential between 5 and 15 ms on each current step and extrapolating back to 0 ms to estimate the voltage drop, and then the voltage drop was plotted against the injected current and fitted with linear regression. The series resistance was subtracted

offline. The input resistance was measured on the hyperpolarizing current injections only, using the voltage minimum within the first 150 ms. The median of Rin values across all steps was taken as the input resistance of the cell. The AP threshold was taken as the point where dV/dT reached 10 V/s, and was measured as the median value of the 1st AP across all depolarizing steps. The AHP was measured as the minimum voltage reached during a 50 ms window after the time of the spike threshold. Only spikes in which the following spike or end of the current step fell outside of this window were included in the analysis. The latency to 1st spike was measured on the maximal current steps. The burst index represents the mean fraction of spikes occurring in bursts, which we defined as two or more spikes with inter-spike intervals of less than 10 ms, and was calculated for each current step amplitude. We then took the highest value from the step amplitudes that induced an average of 5 or more spikes. The Vm variance was calculated in 5 s segments of spontaneous activity, the value for each time point was taken as the median of all the segments in the period before commencing the testing protocol. The sag ratio was calculated as the steady state Vm (median of the final 50 ms) divided by the minimum Vm during the first 150 ms. The value for each cell was taken as the median across all hyperpolarizing current steps.

To examine the evolution of place cell firing rate we re-analyzed data from a previous study, where we made silicon probe recordings from CA1 in mice as they ran back and forth along a virtual linear track (*Bourboulou et al., 2019*). In that study each experiment consisted of up to four sessions, two in a familiar condition followed by one or two in a novel condition. Each session lasted 15–20 min, with 3 min in the dark between sessions. For our analysis we took only data from the second familiar (14 sessions from five mice) and first novel sessions (10 sessions from five mice). The conditions used were the 'object track', 'pattern no object track' and 'pattern object track' (the latter was not described in *Bourboulou et al., 2019*, but consisted of the 'pattern no object track' with identical objects to the 'object track'). Each direction along the track was analyzed independently. To ensure we did not use unstable cells we took only those that had stability and spatial information values above 0.1. The stability was the mean of the distribution of all the pairwise correlations of all the laps. The lap-by-lap firing rate correlation across each session was a Pearson correlation. To calculate the mean firing rates each session was divided into thirds based on the duration of the session, corresponding to the start, middle and end of the session.

Within groups statistical comparisons were made using paired t-tests, and between group comparisons using two-tailed t-tests. Linear regression was used for comparisons between time points of the current-spiking curves. The extracellular firing rates were compared using one-way repeated measures ANOVA. The center of the in-field and out-of-field correlation distributions was tested using either a sign-test or a one-tailed t-test, depending on the normality of the distribution. The threshold for significance was $p=0.05$. Values are given as mean ± SEM unless otherwise stated.

## Acknowledgements

This study was funded by INSERM, Aix-Marseille University, a rising star grant from of the A*MIDEX project (n° ANR-11-IDEX-0001–02) funded by the « Investissements d'Avenir» French Government program (to JE), by the European Research Council under the European Community's Seventh Framework Program (ERC-2013-StG-338141_Intraspace to JE), by the ANR (FLAG-ERA-HBP-partnering project ANR-17-HBPR-0004 HIPPOPLAST) and region PACA. The authors would like to thank Laurent Aniksztejn, Dominique Debanne and Matthew Nolan for comments on the manuscript, Geoffrey Marti for carefully reading and correcting the manuscript, members of the Epsztein lab for discussions and the animal facility, administrative and imaging platforms of INMED for support.

## Additional information

### Funding

| Funder | Grant reference number | Author |
|---|---|---|
| Seventh Framework Programme | ERC-2013-StG 338141 - IntraSpace | Jérôme Epsztein |
| Institut National de la Santé et de la Recherche Médicale | | Jérôme Epsztein |

| Agence Nationale de la Recherche | FLAG-ERA-HBP-partnering project ANR-17-HBPR-0004 HIPPOPLAST | Jérôme Epsztein |
| A*MIDEX | ANR-11-IDEX-0001–02 | Jérôme Epsztein |
| Région Provence-Alpes-Côte d'Azur | | Jérôme Epsztein |

The funders had no role in study design, data collection and interpretation, or the decision to submit the work for publication.

## Author contributions
Peter James Morgan, Conceptualization, Data curation, Software, Formal analysis, Investigation, Visualization, Methodology, Writing—original draft; Romain Bourboulou, Caroline Filippi, Julie Koenig-Gambini, Formal analysis, Investigation; Jérôme Epsztein, Conceptualization, Resources, Supervision, Funding acquisition, Investigation, Project administration, Writing—review and editing

## Author ORCIDs
Jérôme Epsztein (iD) https://orcid.org/0000-0002-5344-3986

## Ethics
Animal experimentation: All experiments were approved by the Institut National de la Santé. et de la Recherche Médicale (INSERM) animal care and use committee and authorized by the Ministère de l'Education Nationale de l'Enseignement Superieur et de la Recherche (agreement number 692 02048.02), in accordance with the European community council directives (2010/63/UE).

## Decision letter and Author response
Decision letter https://doi.org/10.7554/eLife.49915.sa1
Author response https://doi.org/10.7554/eLife.49915.sa2

# Additional files

## Supplementary files
• Transparent reporting form

## Data availability
We provide the source data files used to generate all figures.

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
