## [Decision Letter]

**Acceptance summary:**

This work presents a pioneering study addressing whether theta-burst firing, mimicking the firing of place cells within their place fields, induces homeostatic plasticity of intrinsic excitability in vivo in CA1 pyramidal cells. The authors' results in anesthetized rats suggest that theta-bursts induce a rapid homeostatic decrease in excitability. This reduction in excitability is regulated by an increase Kv1.1 conductance in the axon initial segment. This study indicates that action potential firing can regulate global intrinsic excitability of a neuron independently of dendritic excitability, which may have profound implications for computations performed by CA1 neurons.

**Decision letter after peer review:**

Thank you for submitting your article "Kv1.1 contributes to a rapid homeostatic plasticity of intrinsic excitability in CA1 pyramidal neurons in vivo" for consideration by *eLife*. Your article has been reviewed by two peer reviewers, and the evaluation has been overseen by a Reviewing Editor and Eve Marder as the Senior Editor. The following individual involved in review of your submission has agreed to reveal their identity: Stefan Remy (Reviewer #2).

The reviewers have discussed the reviews with one another and the Reviewing Editor has drafted this decision to help you prepare a revised submission.

Summary:

The study by Morgan et al. presents an interesting and technically challenging set of experiments with potential implications for the regulation of place cell firing by intrinsic plasticity. The authors use current injections via whole-cell patch clamp electrodes in anesthetized rats to induce plasticity of intrinsic excitability in CA1 pyramidal cells. Intrinsic plasticity mechanisms have been reported in acute brain slices and they could be involved in longer lasting regulation of cellular excitability without synaptic input specificity. The presence and potential functional implications of such mechanisms have only rarely been investigated in vivo and thus it is currently unknown whether they are relevant for the regulation of firing rates during behavior. Therefore, this study addresses a novel, potentially relevant topic. A demonstration that intrinsic plasticity mechanisms regulate the firing rates of CA1 place cells during spatial navigation would indeed almost represent a paradigm shift. The authors propose activity-dependent plasticity as a candidate mechanism for preventing the activation of identical cell ensembles within different environments, thereby preventing memory interference.

While the reviewers are convinced by the data that TBS can be evoked in vivo (albeit during anesthesia) and that D-type conductances play a role, they are not yet convinced that such a mechanism would be of major relevance for the regulation of place cell activity and memory mechanisms in response to natural stimuli (and firing patterns) observed during behavior. The additional analyses (as specified below) should provide at least some evidence for or be consistent with the presence of intrinsic plasticity in unperturbed in vivo conditions (place cell firing). Moreover, the limitations of the study should be addressed in the Discussion and the interpretation of the results, together with the expectations generated by Abstract and Introduction, should be toned down.

Essential revisions:

1) The manuscript investigates potential mechanisms of place cell ensemble regulation. We know how place cells fire during behavior from decades of extracellular recordings. Whole-cell work of Albert Lee's group involving the senior author of this study and from the Magee lab even make it possible to simulate the subthreshold and suprathreshold membrane potential changes very precisely. Even if we would assume that a theta-burst pattern very roughly resembles place related firing, multiple repetitions of the same firing within an interval of 15 seconds are extremely unlikely during natural behavior. So it remains unclear to us, how ensemble suppression in different environments could be evoked, if it would exist. If intrinsic plasticity would be evoked by a single TBS or a long constant current injection (1-2s), which would be somewhat more naturalistic, I would be more convinced of its relevance, but it would still remain unclear to me how ensemble suppression could work.

We have one suggestion that does not require additional experimentation, but some analyses, which would increase our confidence in the relevance of such an intrinsic plasticity mechanism for place related firing. It is based on the observation, that CA1 neurons of rodents either on linear tracks or on treadmills quite accurately show repeated activation of place cells in natural firing patterns with repetition intervals that are within the time frame of the induction paradigm. The authors could request such data from a lab that has published such results in the past (they may even have these data available from their own work). In my view, two predictions derive from the current manuscript: a) Repeated activation should result in a gradual decrease of the place cell firing rate and b) The time to first spike should be gradually prolonged.

For the theoretically possible scenario that synaptic and intrinsic plasticity mechanisms counterbalance each other, out of place field firing rates could be analysed, which may also represent input from different cell assemblies and should be gradually depressed. If these predictions are confirmed, it will significantly strengthen the manuscript.

2) A cocktail of anesthetic compounds is used (opiods, benzodiazepines, medetomidine) that generates quite artificial conditions, that are impossible to control for (effects on GABAergic transmission and tonic inhibition, deprivation of excitatory input, unclear effects on neuromodulatory afferents and many unknown effects of the drugs alone and in combination). In these conditions, it is not very surprising that the mechanism observed here (D-type dependent) differs from the mechanism described earlier in slice work (HCN). We agree, that the unperturbed morphology of the neurons in vivo could be a possible explanation, but the authors do not provide any experimental evidence for a role of axonal damage, it is a pure speculation. Not only do the authors fail to reproduce an Ih dependent mechanism (see Campanac et al., 2005; Fan et al., 2005; Narayan and Johnston, 2007), they now present an alternative mechanism. This whole issue should be discussed in a comprehensive and fair way, including a discussion of how well distal dendritic currents (and membrane potential changes) can be resolved in somatic in vivo recordings and the pros and cons of slice vs. in vivo work.

3) Role of inhibition. Although briefly discussed, plasticity of inhibitory transmission and tonic inhibition have not been convincingly ruled out as a potential alternative or additional mechanism. This is important as the use of benzodiazipenes and other drugs acting on GABAergic transmission might have been an important co-factor in the data presented here. The discussion of relevant work is generally a bit too superficial.

---

## [Author Response]

Essential revisions:1) The manuscripts investigates potential mechanisms of place cell ensemble regulation. We know how place cells fire during behavior from decades of extracellular recordings. Whole-cell work of Albert Lee's group involving the senior author of this study and from the Magee lab even make it possible to simulate the subthreshold and suprathreshold membrane potential changes very precisely. Even if we would assume that a theta burst pattern very roughly resembles place related firing, multiple repetitions of the same firing within an interval of 15 seconds are extremely unlikely during natural behavior. So it remains unclear to us, how ensemble suppression in different environments could be evoked, if it would exist. If intrinsic plasticity would be evoked by a single TBS or a long constant current injection (1-2s), which would be somewhat more naturalistic, I would be more convinced of its relevance, but it would still remain unclear to me how ensemble suppression could work.We have one suggestion that does not require additional experimentation, but some analyses, which would increase our confidence in the relevance of such an intrinsic plasticity mechanism for place related firing. It is based on the observation, that CA1 neurons of rodents either on linear tracks or on treadmills quite accurately show repeated activation of place cells in natural firing patterns with repetition intervals that are within the time frame of the induction paradigm. The authors could request such data from a lab that has published such results in the past (they may even have these data available from their own work). In my view, two predictions derive from the current manuscript: a) Repeated activation should result in a gradual decrease of the place cell firing rate and b) The time to first spike should be gradually prolonged.For the theoretically possible scenario that synaptic and intrinsic plasticity mechanisms counterbalance each other, out of place field firing rates could be analysed, which may also represent input from different cell assemblies and should be gradually depressed. If these predictions are confirmed, it will significantly strengthen the manuscript.

We thank the reviewer for suggesting a possible way to test the relevance of our results obtained in anesthetized animals to awake behaving conditions. Following the reviewer’s suggestion, we analyzed data previously obtained in our lab using extracellular silicon probe recordings of CA1 place cells’ activity in head-fixed mice navigating virtual linear tracks in familiar and new conditions. Familiar and new conditions differed by the availability of local visual cues (virtual objects). The pure linear behaviour of mice in these tracks allowed multiple passes through the place field and analysis of the evolution of place cells firing rate over time in both new and familiar conditions. As suggested by the reviewer to search for changes in intrinsic excitability we decided to focus on the out-of-field firing since synaptic plasticity mechanisms are likely to occur inside the place field which might compensate for or even overcome any overall decrease in excitability. We observed a clear and significant tendency for out-of-field firing to decrease over time in the new condition (Start: 1.58 ± 0.11Hz, Middle: 1.47 ± 0.1Hz, End: 1.37 ± 0.1Hz, n = 171, F = 5.17 p = 0.0065, Figure 8A-C), which is consistent with decreased intrinsic excitability over time. And consistent with the presence of synaptic potentiation within the place field we saw an increase in in-field firing rate over time (Start: 3.63 ± 0.19Hz, Middle: 3.92 ± 0.21Hz, End: 4.19 ± 0.22Hz, n = 171, F = 5.02, p = 0.0071, Figure 8E). The changes in in-field firing rate are compatible with synaptic plasticity occurring within the place field (Bittner et al., 2017; Diamantaki et al., 2018) as previously suggested (Mehta et al., 1997). The observed changes in in-field and out-of-field firing rates were experience dependent because they were strongly attenuated or absent in the familiar condition (Figure 8—figure supplement 1). Overall these results are consistent with the presence of an activity-dependent reduction in intrinsic excitability during exploration of a novel environment.

2) A cocktail of anesthetic compounds is used (opiods, benzodiazepines, medetomidine) that generates quite artificial conditions, that are impossible to control for (effects on GABAergic transmission and tonic inhibition, deprivation of excitatory input, unclear effects on neuromodulatory afferents and many unknown effects of the drugs alone and in combination). In these conditions, it is not very surprising that the mechanism observed here (D-type dependent) differs from the mechanism described earlier in slice work (HCN). We agree, that the unperturbed morphology of the neurons in vivo could be a possible explanation, but the authors do not provide any experimental evidence for a role of axonal damage, it is a pure speculation. Not only do the authors fail to reproduce an Ih dependent mechanism (see Campanac et al., 2008; Fan et al., 2005; Narayan and Johnston, 2007), they now present an alternative mechanism. This whole issue should be discussed in a comprehensive and fair way, including a discussion of how well distal dendritic currents (and membrane potential changes) can be resolved in somatic in vivo recordings and the pros and cons of slice vs. in vivo work.

We agree that the anesthetic compounds used in our recordings generate artificial conditions that we cannot ignore when interpreting our data. In the revised version of the manuscript these effects and the fact that they must be considered when interpreting our data is more clearly stated in the Discussion (fifth paragraph). Concerning slice versus in vivo work we would like to highlight that our goal was not to oppose these different preparations which have their advantages and limitations. We are fully aware of the value of previous work using in vitro recordings that directly inspired our own work. In the revised version of the manuscript we tried to account for these different advantages and limitations (see the aforementioned paragraph) and explain our choice of anesthetized in vivo recordings which afford more stability than fully awake preparations at the expense of being in an unnatural brain state. Using this preparation, although we failed to reproduce some forms of Ih plasticity observed in vitro such as a global increase following theta burst firing of CA1 pyramidal cells (Fan et al., 2005; Campanac et al., 2008) we were able to reproduce other forms of Ih plasticity observed in vitro (Figure 5) such as a Ca^2+^- and NMDAR-dependent decrease in Ih following weaker stimulations inducing a lower level of potentiation (Campanac et al., 2008). The complexity of Ih plasticity in vitro and the need to further assess it in vivo in now more clearly stated in the revised Discussion along with possible explanations (Discussion, second paragraph). The fact that assessing the impact of dendritic conductances on somatic excitability via somatic patch-clamp recordings is more limited in vivo is also acknowledged.

3) Role of inhibition. Although briefly discussed, plasticity of inhibitory transmission and tonic inhibition have not been convincingly ruled out as a potential alternative or additional mechanism. This is important as the use of benzodiazipenes and other drugs acting on GABAergic transmission might have been an important co-factor in the data presented here. The discussion of relevant work is generally a bit too superficial.

We agree with the reviewer that inhibition could be involved in the plasticity we describe and have expanded the Discussion to further examine the possibility of plasticity of both tonic (third paragraph) and phasic (fourth paragraph) inhibition. Notably we highlight that burst firing of CA1 pyramidal cells in vitro induces a long-term depression of both somatic and dendritic feedback inhibition (Younts, Chevaleyre and Castillo, 2013) which is unlikely to account for the reduced excitability we observe.